# RUNX Family in Hypoxic Microenvironment and Angiogenesis in Cancers

**DOI:** 10.3390/cells11193098

**Published:** 2022-10-01

**Authors:** You Mie Lee

**Affiliations:** 1Vessel-Organ Interaction Research Center, VOICE (MRC), Kyungpook National University, 80 Daehak-ro, Buk-gu, Daegu 41566, Korea; 2Lab of Molecular Pathophysiology, College of Pharmacy, Kyungpook National University, 80 Daehak-ro, Buk-gu, Daegu 41566, Korea

**Keywords:** RUNX1, RUNX2, RUNX3, hypoxia, angiogenesis, tumor microenvironment, HIF

## Abstract

The tumor microenvironment (TME) is broadly implicated in tumorigenesis, as tumor cells interact with surrounding cells to influence the development and progression of the tumor. Blood vessels are a major component of the TME and are attributed to the creation of a hypoxic microenvironment, which is a common feature of advanced cancers and inflamed premalignant tissues. Runt-related transcription factor (RUNX) proteins, a transcription factor family of developmental master regulators, are involved in vital cellular processes such as differentiation, proliferation, cell lineage specification, and apoptosis. Furthermore, the RUNX family is involved in the regulation of various oncogenic processes and signaling pathways as well as tumor suppressive functions, suggesting that the RUNX family plays a strategic role in tumorigenesis. In this review, we have discussed the relevant findings that describe the crosstalk of the RUNX family with the hypoxic TME and tumor angiogenesis or with their signaling molecules in cancer development and progression.

## 1. Introduction

Emerging evidence has led to the emergence of a novel scenario in which cancer maintenance and expansion are critically regulated by signals from its microenvironment [1]. Tumor progression is strongly influenced by the interaction of cancer cells with their microenvironment, which ultimately determines whether the primary tumor has established eradication, metastasis, or dormant states. A tumor mass is composed of a heterogeneous population of cancer cells as well as various resident and infiltrating host cells, secreted factors, and extracellular matrix proteins, which are collectively known as the tumor microenvironment (TME) [2]. Hypoxia is a well-known form of stress that impairs the biological function of cells and is associated with treatment refractory and anticancer drug resistance in malignant tumors and cancers [3].

The Runt-related transcription factor (RUNX) protein family is evolutionarily conserved and regulates various important developmental and biological processes such as proliferation, differentiation, apoptosis, and cell growth in diverse tissues in a context-dependent manner [4]. There are three RUNX genes (*RUNX1, RUNX2**,* and *RUNX3*) in humans that encode acute myeloid leukemia (AML), the alpha subunit of polyomavirus enhancer-binding protein 2 (PEBP2α), or core-binding factor subunit α (CBFα) [5]. Furthermore, these genes are defined by the presence of a highly conserved 128 amino acid DNA binding/protein–protein interaction domain, called the Runt-homology domain [6]. Additionally, RUNX proteins form a heterodimeric complex with CBFβ, which changes its conformation, exposing its DNA-binding site and increasing its DNA-binding affinity, to exert its function as a sequence-specific trans-activator [7]. Mouse Runx genes are closely related and essential for hematopoiesis, osteogenesis, and neurogenesis, besides being important for other developmental processes [7]. RUNX1 has been associated with leukemia [8,9] and solid tumor development in the skin, lung, intestine, and breast [9,10], while RUNX2 has been associated with osteosarcoma [11,12], thyroid carcinoma [13], as well as breast and prostate cancers [14]. RUNX3 has been associated with gastric cancer [15] and other types of cancers—for example, lung, breast, and pancreatic cancers [16]. As RUNX genes have been found to function as both tumor suppressors and dominant oncogenes in a context-dependent manner, it is imperative to gain further clarity regarding the current experimental evidence associated with their function, especially in hypoxic TMEs and tumor angiogenesis, via comprehensive deliberations.

## 2. Players of Hypoxia and Angiogenesis in Tumor Microenvironment

### 2.1. Players in Hypoxia TME

Hypoxia-inducible factors (HIF-1, -2, and-3) are key transcription factors that regulate the hypoxic cellular microenvironment [17]. Hypoxia is a common phenomenon often associated with poor prognosis that is linked to increased aggressiveness and invasiveness, metastatic behavior, and chemo-resistance in solid tumors [18]. The molecular mechanisms underlying HIF-α stability and activity have been reviewed elsewhere [19,20].

Hypoxia has also been documented to play a role in hematologic malignancies. For instance, HIF-1α is overexpressed in clusters of bone marrow (BM)-resident leukemic cells in pediatric acute lymphoblastic leukemia (ALL) cases but was found to be absent in normal BM [21]. BMs from adult patients with ALL frequently exhibit HIF-1 expression, associated with poor prognosis [22]. Furthermore, BM hypoxia has been reported to promote the dissemination and rehabilitation of multiple myeloma (MM) cells by inducing epithelial mesenchymal transition (EMT)-like features during the progression of MM [23]. Moreover, HIF-1α silencing in MM cells has been shown to inhibit tumor progression due to decreased angiogenesis and bone destruction by the downregulation of proangiogenic and pro-osteogenic cytokines [24]. Consequently, hypoxia has been considered a desirable target for anti-cancer therapy of both solid and liquid tumors.

Other key players in hypoxic responses include histone deacetylases (HDACs) and histone methyltransferases (HMTs), which are crucial for the epigenetic regulation of gene expression. HDACs play a critical role in transcriptional regulation by inducing conformational changes in chromosomal heterochromatin structure via the removal of acetyl groups from ε-amino lysine residues on histone tails. Moreover, it has been well established that the aberrant expression of classical (class I, II, IV) HDACs and alterations in histone acetylation are linked to cancer development and a variety of malignancies, including solid and hematological tumors [25]. Two decades ago, Kim et al. identified that among the HDACs, HDAC1 activity is elevated under hypoxic conditions [26]. HDAC3 is also regulated by hypoxia and HIF-1α-induced HDAC3 is essential for EMT and metastatic phenotype in hypoxic stress conditions [27]. Some types of HDACs, including HDAC1, 4, 5, and 6 have been documented to increase HIF-1α stability by direct deacetylation of HIF-1α or its chaperones, heat shock protein (Hsp)70, and Hsp90 [25].

G9a, a histone methyltransferase (HMT) that targets histone H3, and H3K9 methylation, are usually used as markers of epigenetically silenced genes. G9a HMT has been reported to accumulate in a similar way to HIF-1α under hypoxic conditions [28] and is attributed to the epigenetic silencing of many tumor suppressors involved in the cell cycle, apoptosis, DNA repair, angiogenesis, and metastasis [29]. G9a accumulation under hypoxic conditions has been also shown to increase non-histone targets that regulate the expression of hypoxia-responsive genes [30]. In addition, there is much evidence that G9a HMT is overexpressed and strongly correlated with metastatic cancer progression. Thus, the inhibition or depletion of some HDACs and G9a HMT in experimental systems leads to reduced tumor mass and metastasis, suggesting that they function as oncogenic and metastatic factors [25,29].

### 2.2. Players in Tumor Angiogenesis

Angiogenesis induced by angiogenic factors secreted from cancer cells and TME contributes to the transition of dormant tumors into fast-growing tumors [31]. Angiogenesis is a major phenomenon that is regulated by HIF signaling in response to hypoxic stress to overcome the lack of oxygen and nutrients. Potent angiogenic factors including vascular endothelial growth factor (VEGF) family members are reportedly induced by HIF signaling, which bind to their receptors, VEGFR1-2 and neuropilin [32]. Angiopoietins (Ang-1, -2) and Tie-2 receptors are yet other important molecules induced by hypoxia that are involved in a VEGF pathway-independent signaling pathway. Furthermore, Ang-1 has been reported to be essential for normal vascular development, whereas Ang-2 is mainly expressed in remodeling tissues and the hypoxic TME [33]. Additionally, elevated VEGF and Ang-2 expression is correlated with poor prognosis in various types of tumors [34,35].

Despite being hyper-vascularized, tumor vasculatures are often unable to supply oxygen or efficiently deliver anti-cancer drugs to tumor cells owing to incomplete perfusion through distorted vessels. Consequently, hypoxic conditions within the solid tumor mass can induce the expression of additional angiogenic factors, the immune checkpoint molecule PD-L1 [36], anti-apoptotic factors, and chemo- as well as radio-resistance imparting molecules via HIF signaling [37,38].

## 3. RUNX1 in Hypoxic TME and Angiogenesis

RUNX1 (also called AML1/PEBP2αB/CBFα2) is a key regulator of terminal hematopoiesis during embryonic development and adulthood. Its gene is located on chromosome 21 in humans and was first characterized in acute myeloid leukemia gene 1 (AML1) for t(8;21) translocation in AML cancer patients [39,40]. It is well known for its frequent translocation and mutation in hematological malignancies as reviewed in other papers including this Special Issue [41,42].

RUNX1 is involved in various biological functions such as immune cells, epithelial and epithelial stem cells, hair follicles, and neuronal development [43,44,45]. Consistent with its overarching role, RUNX1 is associated with the pathogenesis of malignancies originating in a variety of tissues, independent of the hematopoietic system, including the breast, ovarian, pituitary, and gastrointestinal systems [46].

Here, we need to look over the relations between the role of hypoxia in the BM microenvironment and leukemia development. Konopleva et al. [47] have reported that hypoxia is a common component of the leukemic BM environment. As such, extensive expansion of hypoxic regions increases with disease progression via labeling the BM of mice bearing xenografts or syngeneic acute leukemia models with the 2-nitroimidazole hypoxia probe pimonidazole [48]. Matsunaga et al. have suggested that hypoxia via HIF-1α might play a role in the maintenance of minimal residual disease in AML [49]. They observed that HIF-1α promoted the arrest of leukemia cells in the intraosseous niche, and speculated that this might contribute to the persistence of residual leukemic cells in AML. In contrast, hypoxia or hypoxia mimetics such as cobalt chloride and desferrioxamine have been reported to stimulate the differentiation of many types of AML cells in vitro [50,51]. A subsequent study has demonstrated that intermittent hypoxia significantly enhances the survival of transplanted leukemic mice via the induction of leukemic cell differentiation in vivo [52]. These reports thus suggest that hypoxia or hypoxia-induced molecules such as HIF-1α induce leukemic cell differentiation. Urged expression of RUNX1 inhibits HIF-1α transcriptional activity and decreased target gene expression, such as VEGF, but that of HIF-1α enhances RUNX1 transcriptional activity, confirming the role of HIF-1α and RUNX1 in angiogenesis and differentiation of leukemia cells [53]. Moreover, the interaction between HIF-1α and Runx1 or HIF-1α and CCAAT/enhancer binding protein alpha (C/EBPα) has been shown to increase the transcriptional activity of Runx1 or C/EBPα. Additionally, HIF-1α reportedly plays a role in all-trans retinoic acid-induced leukemic cell differentiation [54].

The *AML1/ETO* fusion gene is known to induce leukemogenesis in AML cells without mutagenic events. It has been demonstrated that HIF-1α plays a role in the phenomenon of leukemogenesis by interacting with AML1/ETO protein to prime leukemia cells for subsequent aggressive growth. HIF-1α and AML1/ETO fusion proteins thus form a positive regulatory circuit and cooperate to increase DNA hypermethylation through transactivation of the DNMT3a gene [55]. It was further identified that the RNA N^6^-methyladenosine (m^6^A)-reader enzyme YTH N6-methyladenosine RNA-binding protein 2 (YTHDF2) is a key molecule that regulates cell proliferation in t(8;21) AML patients. Chen et al. revealed that YTHDF2 is a target of the AML/ETO-HIF-1α loop and promotes cell proliferation, probably by modulating global m^6^A methylation in t(8;21) AML [56] (Table 1). Therefore, these suggest that in normal HSCs, HIF-1α and RUNX1 interaction is increases RUNX1 activity to induce angiogenesis or HSC differentiation, but in leukemic states, HIF-1α and RUNX1(AML)/ETO fusion protein interaction increase leukemic aggressiveness with proliferating phenotypes.

RUNX1 partner transcriptional co-repressor 1 (RUNX1T1) is a member of the ETO homolog family and is involved in the chromosomal translocation with RUNX1 in AML [57]. It acts as a corepressor by interacting with many transcription factors and recruits nuclear corepressors, such as NCoR, SMRT, and HDACs, resulting in transcriptional repression and dysregulated hematopoietic differentiation during the development of leukemia [58,59]. In human glioma patient tissues, the expression of the RUNX1T1 gene and protein is downregulated, while the expression of HIF-1α is higher than that in normal brain tissues. Moreover, RUNX1T1 reportedly interacts with HIF-1α and recruits proline hydroxylase 2 (PHD2) and GSK3β for HIF-1α degradation, resulting in the inhibition of glioma cells [60] (Table 1). Long noncoding (lnc) RNA RUNX1-IT1 is the intronic transcript 1 of RUNX1, which is also known as chromosome 21 open reading frame 96 (C21orF96). In hepatocellular carcinoma (HCC), it has been identified that lnc RUNX1-IT1 represses HCC cell proliferation, cell cycle progression, and cancer stemness in vitro. Furthermore, its downregulation in HCC samples correlates with an unfavorable prognosis and is mediated by hypoxia-induced HDAC3 in HCC [61].

The fact that hematopoietic cells and ECs have common precursor cells, that is, hemangioblasts [62,63], and that Runx1 is essential for the generation of hematopoietic cells from hemogenic endothelial cells [64] suggests that Runx1 plays an important role in angiogenesis. In physiological angiogenesis, endothelial differentiation and maturation, as well as vascular network formation, are promoted by Runx1 through the expression of VE-cadherin by the repression of insulin-like growth factor-binding protein-3 (IGFBP-3) [65] and Ang1 [66]. Runx1-deficient mouse embryos show angiogenic defects in vital organs with absence of HSCs, suggesting its essential role in the vasculogenesis and angiogenesis [65,67].

In malignant states, the role of RUNX1 in angiogenesis has been explored but the conclusions remain controversial. Runx1 acts as a transcriptional repressor of *VEGF-A* by directly binding to its promoter and has been suggested to have anti-angiogenic activity in AML cells [68] and in a mouse HCC model [69] (Table 2), indicating that RUNX1 performs an anti-angiogenic function by suppressing *VEGF-A* expression. Conditioned media obtained from RUNX1-silenced neuroblastoma cells has been shown to stimulate in vitro tube formation in human umbilical vascular endothelial cells (HUVECs). In addition, the SH-SY5Y and SK-N-SH neuroblastoma cells subjected to RUNX1 knockdown demonstrate increased microvessel density in xenograft tumor tissues [69]. These investigations thus revealed the effect of RUNX1 that is independent of the HIF-1α-mediated hypoxic signaling and hypoxia-induced angiogenesis. However, RUNX1 silencing in U-87 MG human glioblastoma cells has been shown to inhibit tube formation in HUVECs. IL-1β induces expression of RUNX1 in which p38 MAPK pathway is activated for the expression of invasion- and angiogenic molecules, such as MMPs, and VEGF-A [70]. These findings suggest that RUNX1 increases angiogenic function in glioblastoma cells via the IL-1β-RUNX1-p38MAPK-MMPs and VEGF-A signaling axis [70] (Table 2). Together, involvement of RUNX1 in tumor angiogenesis is different depending on the tumor types, and thus, further an in-depth study on the relationship between RUNX1 and various angiogenic factors and their receptors or hypoxia signaling, including HIF-1α, in tumor angiogenesis should be more piloted.

## 4. RUNX2 in Hypoxic TME and Angiogenesis

RUNX2 is a major determinant of osteoblast differentiation and regulates chondrocyte proliferation, differentiation, and hypertrophy during endochondral bone formation [71,72]. A link between Runx2 and HIF-1α in hypertrophic chondrocytes and angiogenesis has been suggested. In wild-type hypertrophic chondrocytes, co-expression of Runx2 and HIF-1α, as well as higher vascular density, are observed, but in *Runx2* knockout mice, the expression of HIF-1α and vascular formation are not observed in growing tibial bones [73] (Table 1). This finding thus suggests a possible role of RUNX2 in angiogenesis in endochondral bone formation. In addition, RUNX2 physically interacts with HIF-1α, and *Runx2* overexpression has been reported to increase HIF-1α levels, although the underlying mechanism has not been identified. Knockdown of *Runx2* is documented to decrease VEGF transcript levels, but it is not essential for the HIF-1α response as hypoxia induces the expression of *VEGF* transcripts in *Runx2*–null cells too [73]. A further study has shown that RUNX2 interacts with the oxygen-dependent degradation domain (ODDD) of HIF-1α and competes with the von Hippel Lindau (pVHL) protein, which is an E3-ubiquitin ligase, resulting in the stabilization of HIF-1α. Furthermore, *RUNX2* overexpression has been documented to increase the angiogenic activity of HUVECs in vitro. This finding may therefore explain why vascular angiogenesis in the hypertrophic zone of the growth plate is mediated by RUNX2 during endochondral bone formation [74]. As HDAC4 has been known to decrease total and acetylated Runx2 [75] and HIF-1α [76] through its deacetylation and transcriptional repressor activities, reduced expression of HDAC4 results in higher levels of Runx2 and HIF-1α, thereby increasing transcription of VEGF and its angiogenic activity on chondrosarcoma cells (Table 1).

Various studies have extensively documented the involvement of RUNX2 in tumor development, progression, and metastasis. Its expression is significantly upregulated in prostate [77], breast [78,79], and colon cancers [80]. *RUNX2* overexpression is associated with an increase in bone metastasis of breast cancers [81] and prostate cancers [82]. *RUNX2* overexpression has been shown to contribute to a more aggressive and metastatic phenotype by altering the expression of many genes involved in migration, invasion, metastasis, apoptosis, and angiogenesis, including VEGF, MMPs, and bone sialoprotein [77,83,84,85] (Table 1). The proangiogenic effects of RUNX2 have been implicated in tumor progression as it has been shown to enhance endothelial cell proliferation, migration, and invasion. In contrast to RUNX1, *VEGFA* expression is induced by RUNX2 [73,86,87]. All these studies consequently suggest that RUNX2 has a proangiogenic role in promoting the early steps of tumorigenesis, and is probably involved in driving bone metastasis in breast and prostate cancers (Table 2).

Extracellular matrix (ECM) stiffness is one of the upstream regulators of *RUNX2* expression. Aberrant angiogenesis has been observed in tumor tissues with an abnormal stiffness of the ECM [88,89]. ECM stiffness represses VEGF secretion, which also involves the expression of RUNX2-mediated-Yes-associated protein (YAP)-serine/arginine splicing factor 1 (SRF1) [90]. Nitric oxide (NO) is yet another regulator of *RUNX2* expression in prostate cancer [91]. NO is a soluble gas produced by nitric oxide synthase (NOS), which has three isoforms: inducible NOS (iNOS), endothelial NOS (eNOS), and neural NOS (nNOS). The control of NO flux exerted by these three synthases has important implications for tumor growth, apoptosis, angiogenesis, and tumor-promoting and anti-tumor effects [92]. In addition, RUNX2 is upregulated under hypoxic stress conditions in cancer cells, resulting in increased expression of the anti-apoptotic gene Bcl-2 [85]. These studies indicate that various features of the tumor microenvironment such as ECM stiffness, NO, and hypoxia, can serve as regulators that increase the expression of *RUNX2* and promote tumor growth by resulting in the expression of genes involved in anti-apoptosis, angiogenesis, and proliferation. Interestingly, glucose has been documented to increase RUNX2 DNA binding activity in endothelial cells (ECs). Glucose has been shown to promote cell cycle progression in both the G2/M and G1 phases in sub-confluent cells. The study also revealed that RUNX2 phosphorylation serves as a mechanism that leads to increased DNA-binding activity of glucose. In addition, glucose has been shown to increase the RUNX2 localization on the *p21* promoter, thereby inhibiting its expression. Thus, inhibition of *RUNX2* expression or its DNA binding under high glucose conditions can be explored as a beneficial strategy to inhibit EC proliferation and angiogenesis in tumors [87,93].

As expected from the role of Runx2 in angiogenesis, RUNX2 is expressed at higher levels in cancers such as osteosarcoma, colon, prostate, and thyroid cancers as well as melanoma [13,94,95,96], suggesting its oncogenic role. Moreover, RUNX2 transactivates genes related to tumor progression, invasion, and metastasis, including survivin, MMP-2, MMP-9, and VEGF [78,80,82,97,98,99,100,101,102]. These findings thus suggest that overexpression of RUNX2 is associated with undesirable outcomes in cancer progression, angiogenesis, and metastasis.

## 5. RUNX3 in Hypoxic TME and Angiogenesis

RUNX3 is associated with multiple developmental functions and the differentiation of immune cells such as CD8 lineage T cells and TrkC-dependent dorsal root ganglion neurons. It is known to function as a tumor suppressor in various carcinomas, including gastric cancer [7]. In this respect, it has been well established that *RUNX3* is silenced in various cancers, and that its silencing is caused by promoter DNA hypermethylation. The silencing of *RUNX3* by hemizygous deletion or DNA hypermethylation and the resultant reduced expression of RUNX3 protein is common in many types of cancers, including bile duct, lung, and pancreatic cancers (Wada et al. 2004; Yanada et al. 2005). For two decades, our research group has been a pioneer in questioning as well as investigating whether the hypoxic tumor microenvironment regulates RUNX3. We have demonstrated that a hypoxic microenvironment can suppress *RUNX3* expression at the transcriptional level via histone modification—that is, methylation and deacetylation via G9a HMT and HDAC1, respectively. Furthermore, *RUNX3* promoter histone methylation and deacetylation have been confirmed in hypoxic microenvironments [103] (Table 1 and Figure 1). It has been identified that G9a HMT is stabilized and accumulated under hypoxia [28,104] and is associated with metastasis and the poor prognosis of multiple human cancers [29,105]. Hypoxic conditions also highly activate HDAC1, which in turn deacetylates H3 histones to decrease the transcription of *pVHL* and *p53* [26]. As the acetylation of RUNX3 protein by p300 [106] and BRD [107] is a key mechanism of its stability and cell-cycle arrest, histone deacetylation plays a major role in RUNX3 inactivation by protein deacetylation. Moreover, as HDAC inhibitors restore *RUNX3* expression and tumor-suppressive function in cancer cells [103,108,109], it has been suggested that under hypoxic conditions, compounds that rescue the epigenetic loss of *RUNX3* expression and protein function could potentially be utilized for the prevention and treatment of various cancers.

In some types of gastric cancer, RUNX3 is regulated at the post-translational level by HMT under hypoxic conditions [111]. The ankyrin repeat domain of G9a, a motif that mediates protein–protein interactions, has been documented to interact with the Runt domain of RUNX3. G9a methylates the Runt domain at lysine residues 129 and 171 of RUNX3, thereby increasing its SMAD-ubiquitination regulatory factor (Smurf)-mediated proteasomal degradation and cytosolic sequestration. The K129R and K171R mutations in *RUNX3* have been shown to suppress its methylation by G9a and recover the RUNX3 activity under hypoxic conditions. The methylation of RUNX3 by G9a inhibits the binding of p300 and CBFβ, and then RUNX3 transactivation activity for target gene expression. Furthermore, chromatin immunoprecipitation sequencing (ChIP-seq) and DNA microarray analysis of K129R and K171R mutants under hypoxic conditions have revealed the involvement of novel RUNX3 target genes in the inflammatory response, leukocyte chemotaxis, and cell death [111] (Table 1 and Figure 1). Therefore, investigations into the roles of these novel RUNX3 target genes have the potential to advance the current understanding of the RUNX3-mediated cellular responses in the future.

Regarding the role of RUNX3 in angiogenesis, it is interesting to note that hypoxia-induced HIF-1α is destabilized by RUNX3. A mechanistic study has demonstrated that RUNX3 interacts with the HIF-1α C-terminal activation domain as well as PHDs, which bind to and hydroxylate proline residues in HIF-1α ODDD. These results suggest that RUNX3 can bind to PHD1 or PHD2 and then recruit HIF-1α ODDD for the hydroxylation of HIF-1α for further proteasomal degradation [110]. As a result, RUNX3 may be considered to be essential for HIF-1α degradation, and the downregulation of RUNX3 may be a possible pathway to increase HIF-1α stability under hypoxic conditions. In addition, RUNX3 overexpression significantly inhibits hypoxia-induced angiogenesis, and siRNA against PHD2 has been documented to restore the RUNX3-mediated inhibition of angiogenesis (Table 1, 2 and Figure 1), suggesting that the interaction between RUNX3 and PHD is significant for the degradation of HIF-1α [110].

Hypoxia-induced microRNAs (miRNAs) have been reported to regulate angiogenesis, apoptosis and cell proliferation in gastric cancer cells through the targeting of RUNX3. For instance, miR-130a and miR-495 are upregulated under hypoxic conditions and bind to the RUNX3 3′-untranslated region (3′-UTR) to target RUNX3 mRNA translation in gastric cancer cells. Furthermore, the combination of miR-130a and miR-495 significantly inhibited the expression of RUNX3 as well as its target genes such as *p21* and *Bim*. In addition, the antagomiRs specific for miR-130a and miR-495 significantly reduced angiogenesis in vivo [112] (Table 1 and 2). This report thus suggests that miR-130a and miR-495 could be further explored as potential therapeutic targets for the recovery of RUNX3 expression under hypoxic conditions. Taken together, these results suggest a novel and critical role for RUNX3 in hypoxic responses such as cancer progression, angiogenesis, stem cell maintenance, and ischemic diseases.

RUNX3 expression is negatively correlated with VEGF expression and microvascular density in the tissues, suggesting its anti-angiogenic role in human gastric cancers [113]. RUNX3 transcriptionally inhibits the expression of VEGF through binding to putative RUNX3-binding sites of the *VEGF* promoter directly [113]. Its anti-angiogenic activity was confirmed by the decreased expression of VEGF and von Willebrand factor (vWF) in RUNX3-overexpressed human microvascular endothelial cells (HMECs) [114]. Like RUNX1, RUNX3 is prominently expressed in hematopoietic cells and different subsets of neurons [115,116]. RUNX3 is specifically expressed in CD34+ HSCs and several hematopoietic cell lines, both normal and malignant in human [116,117]. CD34+ HSCs differentiate into endothelial progenitor cells (EPCs), ancestors of endothelial cells, and EPC differentiation is suppressed by the haploinsufficiency of Runx3 through HIF-1α [118] (Table 2). The function of RUNX3 in hematopoiesis and EPC differentiation leads us to expect that it controls the inflammation responses by the regulation of inflammatory cells [1] and vascular endothelial cells in TME, as well as hematological malignancies [16,20].

## 6. Conclusions

The role of RUNX proteins in the inflammation and hypoxic stress is critical for initial tumor formation and early tumorigenesis because inflammation is suggested to be a putative initiator of carcinogenesis [119] and substantial hypoxia is induced in inflammatory lesions [120]. All RUNX family proteins contain a conserved Runt homology domain but tissue-specific and context-dependent regulatory mechanisms and functions. Furthermore, the RUNX family proteins play a unique role in the regulation of vascular ECs identity and tumor angiogenesis. RUNX1 acts as an anti-angiogenic factor by inhibiting HIF-1α activity as well as an inhibitor of VEGF gene expression by binding its promoter. However, RUNX1/ETO fusion protein interacts with HIF-1α for tumor progression. RUNX2 stabilizes HIF-1α protein for regulating optimal angiogenesis in chondrocytes, but overexpressed RUNX2 in cases of cancer increases VEGF expression and anti-apoptotic activity under hypoxia, resulting in increased tumor progression. RUNX3 is silenced by hypoxia both at the gene and protein levels in cancers by hypoxia-induced HDAC1 and/or G9a HMT. RUNX3 plays an anti-angiogenic role through the degradation of HIF-1α under hypoxia or the direct suppression of VEGF gene expression in cancers. Consequently, it can function in the normalization of abnormalities of tumor vasculature in vascular ECs as well as in hypoxic TME through distinct means of action. Therefore, the manipulation of Runx-family proteins in a hypoxic microenvironment and vascular ECs through the more precise delineation of regulation pathways could be utilized in future research and the development of cancer therapeutics.

## Figures and Tables

**Figure 1 cells-11-03098-f001:**
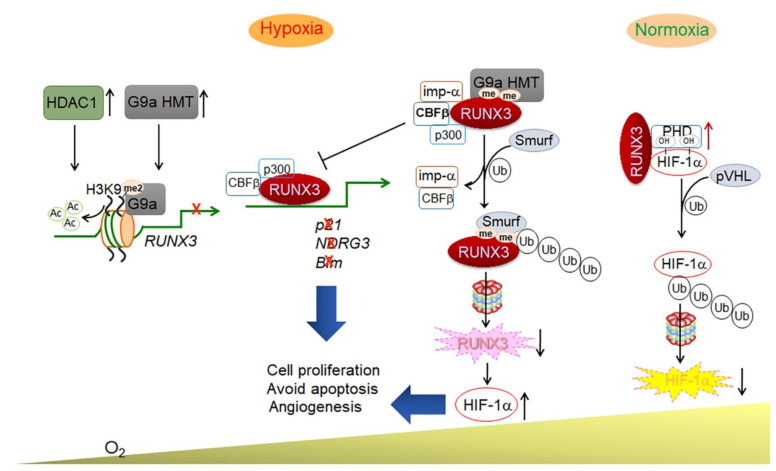
Regulation of RUNX3 on HIF-1α and cellular responses under hypoxic TME. RUNX3 interacts with PHDs and the HIF-1α C-terminal activation domain and induces hydroxylation (OH) of HIF-1α by PHDs, resulting in HIF-1α degradation under hypoxic TMEs and/or normoxic conditions [110]. Hypoxia-induced HDAC1 and G9a HMT increase histone deacetylation and H3K9 dimethylation on *RUNX3* promoter to inhibit its gene transcription [103]. Hyperactive G9a HMT methylates non-histone protein RUNX3 at lysine 129 and 171 residues (me) on the Runt domain and increases its Smurf-mediated ubiquitination and proteasomal degradation under hypoxic TME. In turn, the decreased RUNX3 levels result in the stabilization of HIF-1α. In addition, G9a-mediated methylation of RUNX3 inhibits CBFβ- and p300-mediated transactivation activity of RUNX3. Thus, decreased RUNX3 protein level and activity inhibits the expression of RUNX3 target genes but increases the expression of HIF-1α target genes involved in cell proliferation, apoptosis, and angiogenesis. Importin α7 (imp-α) mediates RUNX3 cytosolic translocation from the nucleus after methylation by G9a HMT [111].

**Table 1 cells-11-03098-t001:** Summary of RUNX functions in hypoxic TMEs.

Protein	Regulation	Phenotype	Experimental Systems
RUNX1	HIF-1α interaction w/RUNX1/ETO → DNA hypermethylation by DNMT3a transactivation. Target: YTHDF2	▪Increase in cancer cell proliferation and aggressiveness	human AML cells & mouse leukemia model, human t(8;21)AML patient
	HIF-1α interaction w/RUNX1T1 → Recruitment of PHD1/GSK3β to HIF-1α for HIF-1α degradation	▪Inhibition of cancer cell proliferation and invasion	Glioma cells
	HIF-1α interaction w/Runx1 at Runt domain → Transcriptional activity of HIF-1α → Transcription activity of Runx1	▪Decreased HIF-1α target gene expression, VEGF Activation of Runx1 activity	In vitro overexpression of Runx1 or HIF-1α in leukemia cells
RUNX2	Interaction w/ODDD of HIF-1α	▪HIF-1α stabilization	ATDC5 chondrocytes HEK293 cells in vitro, Runx2 KO mice
	Direct interaction w/HIF-1α	▪Increased HIF-1α	Hypoxic C3H10T1/2 mesenchymal cells, MC3T3-E1 preosteoblast cells
	Hypoxia→ HDAC4→ Deacetylation of Runx2 and HIF-1α	▪Repressed activity of RUNX2 and HIF-1α	Chondrosarcoma, pVHL-null kidney cancer cells
	RUNX2 o/e	▪Apoptosis resistance	Hypoxic LNCaP prostate cancer cells
RUNX3	Histone modification by HDAC1 & G9a HMT at the promoter	▪RUNX3 gene silencing	Hypoxic conditions w/human gastric cancer cells
	Interaction w/PHDs & HIF-1α	▪HIF-1α degradation, decreased HIF-1α target genes	HEK293, human gastric cancer cells
	RUNX3 K129 & K171 methylation by G9a HMT	▪RUNX3 degradation, decreased transactivation activity, increased tumor growth w/decreased tumor cell apoptosis	Hypoxic conditions w/human gastric cancer cells, mouse xenograft model
	Hypoxia-induced miR-130a, miR-495 target RUNX3 mRNA	▪Decreased RUNX3 translation	Hypoxic conditions w/human gastric cancer cells

**Table 2 cells-11-03098-t002:** Summary of RUNX functions in tumor angiogenesis.

Protein	Regulation	Phenotypes	Experimental Systems
RUNX1	Direct binding to *VEGF-A* gene promoter →Repression of *VEGF-A* gene expression	▪Suppression of VEGF protein secretion ▪Decreased HCC cell proliferation, migration, and tumor growth	AML cells HCC cells and mice model
	RUNX1 silencing	▪Increase tube formation	-SiRNA transfection and conditioned media (CM) from neuroblastoma cells
		▪Increase microvessel density -inhibit EC tube formation	-Mouse xenograft neuroblastoma model -SiRNA transfection and (CM) from U-87 MG human glioblastoma cells
	IL-1β →increase RUNX1 via p38 MAPK	▪Increase MMPs and VEGF-A ▪Increase EC tube formation, cancer cell migration	SiRNA transfection and (CM) from U-87 MG human glioblastoma cells, HUVEC
RUNX2	IGFIR-mediation	▪EC tube formation	IGF-induced expression of Runx2 in HBMEC
	Direct increase VEGF transcription	▪VEGF mRNA expression	Hypoxic C3H10T1/2 mesenchymal cells, MC3T3-E1 preosteoblast cells
	RUNX2 o/e	▪Increased VEGF, worse prognosis ▪Increased VEGF, MMPs osteopontin & worse prognosis	Human breast cancer specimens Human prostate cancer cells (LNCaP, PC3)
RUNX3	Direct binding to *VEGF-A* gene promoter	▪Suppression of VEGF-A level, tumor microvessel density, tumor growth & metastasis	Gastric cancer cells, xenograft mouse model, human gastric cancer specimens
	Interaction w/PHDs & HIF-1α	▪Anti-angiogenic	CM of human gastric cancer under hypoxia, HUVEC tube formation, migration, proliferation, CAM assay, Matrigel plug assay
	AntagomiR-130a, antagomiR-495 recover RUNX3 protein level	▪Anti-angiogenic	CM of human gastric cancer Matrigel plug assay
	HIF-1α inhibition	▪Suppressed EPC differentiation	Runx3 heterozygote mouse

## Data Availability

Not applicable.

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
