# Peer review of "RUNX Family in Hypoxic Microenvironment and Angiogenesis in Cancers"

_cells, 2022, doi:10.3390/cells11193098_

Round 1

Reviewer 1 Report

The article entitled “RUNX family in hypoxic microenvironment and angiogenesis in cancers” is a comprehensive review highlighting the critical the role and modulation of RUNX family proteins in hypoxic stress and angiogenesis. Tables 1 and 2 present a detailed summary on the theme proposed in the manuscript, and the figure is easy to understand and complement the text, which would help the reader to understand. I do not have any major concerns; however, it would be nice to shorten the introduction about hypoxia and angiogenesis to focus more on the function of RUNX family on hypoxia and angiogenesis and also a better summary with an insight for therapeutic implications would improve the quality of the review.

Author Response

Responses: As suggested the reviewer, we shortened the introduction about hypoxia and angiogenesis found in many other excellent review papers.

We rewrote the conclusion of this review as a summary as the reviewer suggested as follows,

Conclusions

The role of RUNX proteins in the inflammation and hypoxic stress is critical for initial tumor formation and early tumorigenesis, because inflammation is suggested to be a putative initiator of carcinogenesis [120] and substantial hypoxia is induced in inflammatory lesions [121]. All RUNX family proteins contain a conserved Runt homology domain but tissue-specific and context-dependent regulatory mechanisms and functions. Furthermore, the RUNX family proteins play a unique role in the regulation of vascular ECs identity and tumor angiogenesis. RUNX1 act as an anti-angiogenic factor by inhibiting HIF-1α activity as well as an inhibitor of VEGF gene expression by binding its promoter. But RUNX1/ETO fusion protein interacts with HIF-1α for tumor progression. RUNX2 stabilizes HIF-1α protein for regulating optimal angiogenesis in chondrocytes, but overexpressed RUNX2 in cancer increases VEGF expression and anti-apoptotic activity under hypoxia, resulting in increased tumor progression. RUNX3 is silenced by hypoxia both at the gene and protein levels in cancers by hypoxia-induced HDAC1 and/or G9a HMT. RUNX3 plays an anti-angiogenic role by the degradation of HIF-1α under hypoxia or direct suppression of VEGF gene expression in cancers. Consequently, they can function in the normalization or abnormalities of tumor vasculature in vascular ECs as well as in hypoxic TME in distinct ways of action. Therefore, manipulation of Runx-family proteins in a hypoxic microenvironment and vascular ECs through more precise delineation for regulation pathways would be utilized in promising research directions and development of cancer therapeutics.”

Reviewer 2 Report

The manuscript presented for review describes the involvement of the RUNX family in the tumor microenvironment, hypoxia, and angiogenesis. The article is clear, well-structured manner. Tables and figure are clear and they are also a good summary. The cited references is predominantly older than 5 years, although it seems necessary due to the adequacy of the subject / information in the cited references. Therefore, this article should be published after a few minor corrections: - section introduction, lines 43-46 - the sentence requires redrafting, in its present form is incomprehensible as if something is missing, - section introduction, line 58 - there is information that RUNX in other types of cancers ..... can the authors mention a few examples as for previous RUNX members - page 4 line 166 - wrong font

Author Response

Responses: 1. As for lines 43-46, we rephrased the second one as follows, “There are three RUNX genes (RUNX1, RUNX2, and RUNX3) in humans that encode acute myeloid leukemia (AML), or the alpha subunit of polyomavirus enhancer-binding protein 2 (PEBP2α), or core-binding factor subunit α (CBFα) (Blyth et al. 2005).”

  1. As for line 58-, we added the examples as other cancers as follows, “for example, lung, breast, and pancreatic cancers (Manandhar and Lee 2018)”.
  2. We corrected its wrong font.

Reviewer 3 Report

MS RUNX family in hypoxic microenvironment and angiogenesis in cancers

Type of the article: a review.

Topic: Role of RUNX family in angiogenesis and cancer.

Interest: moderate to high, particularly for those interested in basic research, angiogenesis, and cancer development and progression.

Novelty. Few articles were published on this topic, so the current review is welcome.

Abstract. Well-written includes some general and enough information about the content of the review.

Comments. In the first subchapter, the author shows the involvement of RUNX family in cancer. As RUNX 1,2,3 are involved in a broad spectrum of cancers, this review is welcome. Hypoxia and cell and tissue changes that are induced are described in various types of cancer, including the possible mechanism of action. “…tumor progression is accompanied 124 by angiogenesis (Fukumura et al. 2018; Hanahan and Weinberg 2011; Jain 2014). This was demonstrated long before 2011m and the remarkable publication of Hanahan and Weinberg is a review. It is indicated to show primary sources, back to ’90. The first phrase from 2.2.: it is better to cite directly Folkman et al which really demonstrated the transition to a vascularized tumor. Data on the relationships between the RUNX family, HIF, histones, and DNA hypermethylation, and their effects on cancer cells are summarized in Table 1, very useful to the reader. There are a lot of data that were shown, discussed, and analyzed, and because of this reason, the author should consider a short chapter on future direction and perspectives, Taken together, it is a well-written review, useful for the reader.

Formal aspects. The paper is written according to the requirements of the journal.

Language and grammar. OK.

Figures. Just one drawing, OK.

Tables. OK.

References. Konopleva et al (l.166) are not included in the reference list, and in the text, the year of publication is not given. The list includes 124 titles written according to the requirements of the journal.

To the Editor. I strongly recommend the publication of this review, with minor changes.

Author Response

Reviewer 3.

In the first subchapter, the author shows the involvement of RUNX family in cancer. As RUNX 1,2,3 are involved in a broad spectrum of cancers, this review is welcome. Hypoxia and cell and tissue changes that are induced are described in various types of cancer, including the possible mechanism of action. “…tumor progression is accompanied 124 by angiogenesis (Fukumura et al. 2018; Hanahan and Weinberg 2011; Jain 2014). This was demonstrated long before 2011m and the remarkable publication of Hanahan and Weinberg is a review. It is indicated to show primary sources, back to ’90. The first phrase from 2.2.: it is better to cite directly Folkman et al which really demonstrated the transition to a vascularized tumor. Data on the relationships between the RUNX family, HIF, histones, and DNA hypermethylation, and their effects on cancer cells are summarized in Table 1, very useful to the reader. There are a lot of data that were shown, discussed, and analyzed, and because of this reason, the author should consider a short chapter on future direction and perspectives, Taken together, it is a well-written review, useful for the reader.

Konopleva et al (l.166) are not included in the reference list, and in the text, the year of publication is not given. The list includes 124 titles written according to the requirements of the journal.

Responses: We added the mentioned reference.
